# Dual-Wavelength Photoacoustic Computed Tomography with Piezoelectric Ring-Array Transducer for Imaging of Indocyanine Green Liposomes Aggregation in Tumors

**DOI:** 10.3390/mi13060946

**Published:** 2022-06-15

**Authors:** Xin Sun, Han Shan, Qibo Lin, Ziyan Chen, Dongxu Liu, Zhankai Liu, Kuan Peng, Zeyu Chen

**Affiliations:** 1State Key Laboratory of High Performance Complex Manufacturing, College of Mechanical and Electrical Engineering, Central South University, Changsha 410083, China; 203712182@csu.edu.cn (X.S.); hanshan@csu.edu.cn (H.S.); qlin003@csu.edu.cn (Q.L.); c_zy100@163.com (Z.C.); 2ULSO (Shanghai) Tech Co., Ltd., Shanghai 201821, China; info@ulsotech.com (D.L.); ulsotech@163.com (Z.L.); 3Department of Biomedical Engineering, School of Basic Medical Science, Central South University, Changsha 410083, China

**Keywords:** photoacoustic computed tomography, piezoelectric transducer, indocyanine green, liposome, guidance

## Abstract

Recently, indocyanine green (ICG), as an FDA-approved dye, has been widely used for phototherapy. It is essential to obtain information on the migration and aggregation of ICG in deep tissues. However, existing fluorescence imaging platforms are not able to obtain the structural information of the tissues. Here, we prepared ICG liposomes (ICG-Lips) and built a dual-wavelength photoacoustic computed tomography (PACT) system with piezoelectric ring-array transducer to image the aggregation of ICG-Lips in tumors to guide phototherapy. Visible 780 nm light excited the photoacoustic (PA) effects of the ICG-Lips and near-infrared 1064 nm light provided the imaging of the surrounding tissues. The aggregation of ICG-Lips within the tumor and the surrounding tissues was visualized by PACT in real time. This work indicates that PACT with piezoelectric ring-array transducer has great potential in the real-time monitoring of in vivo drug distribution.

## 1. Introduction

Cancer is a common public disease and has a high lethality rate [1]. It is critical to develop a non-invasive imaging method for imaging the structural information of tumors and the in vivo distribution of drugs. Photoacoustic (PA) imaging is a new biomedical non-invasive imaging modality due to its especially hybrid imaging characteristics [2]. It is based on the ultrasound wave that is excited by short pulse lasers and detected by the transducer. PA imaging combines the advantages of higher spatial ultrasonic resolution and richer optical contrast to provide tissue information with endogenous or exogenous agents, such as oxyhemoglobin (HbO_2_), deoxyhemoglobin (Hb), gold nanoparticles, and indocyanine green (ICG) [3,4,5,6]. Compared to ultrasound imaging, magnetic resonance imaging (MRI), X-ray computed tomography (X-ray CT), positron emission tomography (PET), and pure optical imaging (fluorescence imaging) [7,8], PA imaging has shown its unique advantages including higher spatial resolution, lower cost, richer optical contrast, and no radiation [9].

In the PA imaging system, there are several common types of ultrasound transducers [10], such as single-element transducers [11], handheld linear transducers [12], and curve-array or ring-array transducers [7,13,14,15]. Photoacoustic computed tomography (PACT), as a common form of PA imaging, has shown the excellent potential in clinical applications, including monitoring the circulating tumor cells [16], whole-body imaging for small animals [14], imaging for cancer and inflammation [13,14], therapy monitoring and molecular imaging [17]. To date, a variety of piezoelectric transducers with different geometries have been used in PACT systems, especially ring-array transducers [7,14,15,18,19]. Ring-array transducers can achieve high spatial resolution and rich optical contrast in deep tissues, due to their wide field of view (FOV) [14,20]. Thus, it is essential to develop a ring-array-based PACT system for biomedical imaging (e.g., molecular imaging) [6,21]. 

Nowadays, a wide range of PA contrast agents with excellent photothermal characteristics [6], such as indocyanine green (ICG) and gold nanoparticles [22,23,24], have been used for cancer treatment [25]. ICG, as a near-infrared (NIR) dye approved by the U.S. Food and Drug Administration (FDA) [26], has attracted tremendous attention in photothermal therapy, due to its phototherapeutic capability. However, the clinical use of ICG is limited because it is unstable in water and can be rapidly cleared in the liver. ICG liposomes (ICG-Lips) overcome these limitations with the advantages of improved solubility of ICG and prevention of degradation during administration [27,28]. To improve the photothermal effects of ICG-Lips, laser irradiation should be conducted according to the accumulation of ICG-Lips in the tumor. Therefore, a high-sensitivity PA imaging system for monitoring the dynamic distribution of ICG-Lips is desirable. 

In this paper, we developed a dual-wavelength PACT system with the piezoelectric ring-array transducer to detect the ICG-Lips and assessed their distribution in the tumor. The engineered PACT system consisted of a 512-element ring-array transducer, an OPO laser, and a Verasonics data acquisition system. A 780-nm laser beam and a 1064-nm laser beam were used to obtain the in vivo distribution of ICG-Lips and the structural information of tumor tissues simultaneously. The results indicated that the PACT system enabled the real-time monitoring of ICG-Lips accumulation in tumors. Due to the non-invasiveness and high spatial resolution, our PACT system provides a convenient way to image ICG-Lips in tumors in real time and can be used as a promising imaging platform for phototherapeutic guidance.

## 2. Materials and Methods

### 2.1. Materials

Dipalmitoyl phosphatidylcholine (DPPC), cholesterol, and 1,2-distearoyl-sn-glycero-3-phosphoethanolamine-N-[amino(polyethylene glycol)2000] (DSPE-PEG-2000) were purchased from Xi’an Ruixi Biological Technology Co., Ltd., Xi’an, China.

### 2.2. System Configuration

The PACT system is shown in Figure 1a. The system was controlled by the data acquisition system (Verasonics, Inc., Washington, USA, 256-channel). A trigger (1 µs pulse duration, 20 Hz), outputted from the data acquisition system, was received by DG645 digital delay/pulse generator (ThinkSRS, Inc., CA, USA). DG645 was triggered by falling edge and generated two triggers with a difference of 216 µs (Flash lamp (FL) trigger and Pockels cell (PC) trigger, 100 µs pulse duration, 20 Hz, Figure 1c). The SpitLight 600 OPO laser (Innolas Laser GmbH, 10 ns pulse duration, 20 Hz), triggered by FL trigger and PC trigger, was used as the irradiation source. For the tissues and ICG-Lips imaging, a 1064 nm laser beam and a 780 nm laser beam were transmitted separately through the optic-fiber light delivery. The light intensity at the wavelength of 1064 nm and 780 nm were ~ 7.38 mJ/cm^2^ and ~17.2 mJ/cm^2^, respectively, which was below the American National Standards Institute (ANSI) safety. To homogenize light spots, the eight-branch fiber was then placed evenly around the center of the ring-array transducer (ULSO, Inc., Shanghai, China) (Figure 1b). 

We used a customed 512-element ring-array transducer for the PA signal detection. The transducer consisted of such four arc-shaped parts that they could form a complete ring-array with a diameter of 100 mm. 

Due to the limited channels of the Verasonics system, we divided the ring-array into two parts with 2:1 multiplexing. Two laser firings were employed to form a complete 512-channel acquisition. The data acquisition system obtained the PA signal at a sampling rate of 62.5 MHz and processed the data with MATLAB (R2020a). At the same time, data transfer for imaging was operated at a rate of 4 frames/sec in real time.

### 2.3. Design and Fabrication of Piezoelectric Ring-Array Transducer

A piezoelectric ring-array transducer (diameter, 100 mm) was proposed to achieve a self-focusing FOV. First, piezoelectric composites were split into multiple periodic arrays by the cutter with a kerf of 0.055 mm. The pitch between the elements was 0.618 mm. To achieve a self-focusing anxiety of 50 mm, the piezoelectric composite was machined into an arc (with a radius of 50 mm) in the elevation direction. The thickness of the piezoelectric composite was 0.3 mm, which was determined using the designed central frequency and sound velocity of the material. Next, the backing material was filled in the kerf between the array elements and behind the piezoelectric layer. The thickness of the acoustic matching layer was 0.13 mm. The configuration of the ring-array transducer is shown in Figure 2a. The ring-array transducer was composed of a matching layer, a backing layer, a piezoelectric composite, and an ABS case. Figure 2b shows the photograph of the piezoelectric ring-array transducer. The design parameters of the ring-array transducer are shown in Table 1. The properties of the piezoelectric layer are shown in Table 2. 

### 2.4. Transducer Characterization

For the basic parameter determination of the transducer, a test block (a ball with a diameter of 25 mm) was placed in the FOV and the impulse response waveforms of the ring-array transducer elements were recorded by the oscilloscope (Keysight, DSO2012A). In the measurements, electrical pulse was generated from a pulser/receiver (JSR, DPR300). The time-domain waveforms and their corresponding frequency spectra were detected. The transmission responses of the 25-th and 96-th elements are shown in Figure 3. The center frequency of the ring-array transducer was 5 ± 0.5 MHz, and the −6 dB bandwidth was greater than 55%. 

Resolution and the FOV play a crucial role in the imaging quality [29]. For the characterization of the transducer, we measured the resolution and the FOV with ultrasound mode. To measure the lateral resolution, a short section tungsten filament of 100 µm diameter was placed vertically in the FOV. Radial scan movements had an increment of 0.5 mm. Figure 4a shows the trend of linear decrease in lateral resolution. The resolution decreases as the tungsten filament moves away from the center of the transducer. 

The focused zone of the FOV was shaped like a disk. For FOV measurements, the tungsten filament was placed across in radial section of the ring-array. The step of radial movement was 0.25 mm and the elevation direction movement step was 0.1 mm. Figure 4b shows the cross-section passing through the center of the ring-array and cutting the ring-array into two semicircles. The full width at half maximum (FWHM) of the cross-section FOV focus area was 13 mm (Figure 4c). The FWHM was 0.6 mm in the center area (x = 0.0 mm) of the elevation direction, and 0.7 mm at the edge (x = 6.5 mm) of the focus area (Figure 4d). 

### 2.5. Geometry Correction of the Transducer

Since the ring-array was composed of four circular arc-shaped parts, there would be geometric errors in the assembly process. We should correct this error in the image reconstruction process. We drew inspiration from the work of Lin et al. [7]. We placed a tungsten filament (~100 µm) vertically in the FOV to excite the PA effect as a point source (x_p’_, y_p’_) signal. In the image reconstruction, we assumed that the coordinates of the array elements were arranged uniformly in a circle. Figure 5a shows the definition of both the array element coordinates and the FOV of the image reconstruction area. (x_n_, y_n_) represents the position of the n-th array element and (x_p_, y_p_) represents the pixel location of the point source. L is the size of the image reconstruction area. L_p,n_ is the distance between the array element (x_n_, y_n_) and the pixel point (x_p_, y_p_). In fact, it is very difficult to obtain the coordinates of the point source (x_p’_, y_p’_), but we can find the pixel point (x_p_, y_p_) corresponding to the point source (x_p’_, y_p’_) in the image reconstructed region. We divided the reconstructed area into a number of pixel points with a size of 100 µm. The distance of each pixel point to the array of elements can be easily achieved from the equation:(x_n_ − x_p_)^2^ + (y_n_ − y_p_)^2^ = L_p,n_n = 1, 2, 3, …, 512(1)

Furthermore, we fitted the point source signals collected by each element into a distance curve in MATLAB. By comparing the L_p,n_ with the distance curve, the exact location of the pixel point (x_p_, y_p_) can be found. 

Due to the acquired signal of elements detection being a time domain signal, we can measure the propagation time of PA signal from a point source (x_p’_, y_p’_) to n-th element t_p’,n_ precisely. The propagation time of PA signal from pixel point (x_p_, y_p_) to n-th element t_p,n_ can be measured from the fellow equation:L_p,n_ = t_p,n_ × v n = 1, 2, 3, …, 512(2)
v, representing the speed of sound in water, is 1.50 mm/µs. Thus, we can obtain the correction delay T_n_ for each array element from the difference between t_p’,n_ and t_p,n_. A 10 × 10 mm sample of leaf vein was used to verify the correction effect (Figure 5b). Figure 5c shows the leaf vein images without delay correction. The corrected image is shown in Figure 5d. The enlarged details in the bottom left of the figure show the splitting artifacts caused by the uncorrected geometry errors of the transducer.

### 2.6. ICG-Lips Preparation

Figure 6a shows the process of the ICG-Lips. The ICG-containing liposomes were prepared using a thin-film hydration method. Briefly, DPPC, DSPE-PEG-2000, and cholesterol at a molar ratio of 10:1:3 were dissolved in 5 mL chloroform. The solution was evaporated under reduced pressure for at least 4 h to completely remove the solvent and obtain the lipid thin film. Then, 4 mL ultrapure water containing ICG at a concentration of 1 mg/mL was added to the flask. The lipid film was hydrated with ICG solution at 50 °C for 30 min. After hydration, the suspension was sonicated for 20 min and then extruded 15 times using an extruder with a 100 nm membrane. Finally, the prepared ICG-Lips were purified by dialysis.

### 2.7. Characterization of the ICG-Lips

The transmission electron microscope (TEM) image of ICG-Lips is shown in Figure 6b. The ICG-Lips showed a spherical morphology with a small size. The Z-average value of ICG-Lips measured by dynamic light scattering (DLS) was 129±49 nm (Figure 6c). The particle size determined by DLS was consistent with the size determined by TEM. To guide PACT imaging, the UV-Vis spectra of ICG and ICG-Lips were measured. The absorption peak of ICG and ICG-Lips in phosphate-buffered saline was determined at 780 nm (Figure 6d). 

### 2.8. Animal Preparation

All procedures involving animal experiments were approved by the animal ethics committee of Central South University, China. 

4−5 weeks old BALB/cA mice (15−20 g body weight, male) were purchased from Hunan SJA Laboratory Animal Co., Ltd. (Hunan, China) and used for experiments. To reduce the effect of respiration, the mice were anesthetized with 1% isoflurane using a compact small animal anesthesia machine (RWD, R500) and placed on a custom-made hollow acrylic plate. The function of the hollow acrylic plate was to support the mouse while reducing the reflection of PA signals. For tumor inoculation, 2 × 10^7^ Lewis lung carcinoma (LLC) cells were subcutaneously implanted in the back of the mouse. Tumor-bearing mice were incubated for two weeks. 

### 2.9. Data Process

The data acquisitions with 780 nm and 1064 nm laser wavelengths were performed to image the ICG-Lips and the mouse tissues respectively. All data processing was performed in MATLAB. Then, the data were reconstructed with the conventional back-projection method [30]. In this work, we selected the tumor part in the 780 nm laser wavelength image and the surrounding tissues in the 1064 nm image as the region of interest (ROI). Firstly, we started with the hue, saturation, value (HSV) color mode for imaging with the colormap of hot and gray separately. Then, we converted the processed HSV image to the RGB color space and displayed the new RGB images for the fusion of the 780 nm image and 1064 nm image. For fused images, the pixel values in the RGB color space were calculated using the following equation [31]: R, G, B = (1− M) **·** × A_1064_ + M **·** × A_780_
(3)

Here, R, G, and B represent the red, green, and blue three-color spaces of the pixel. A_780_ and A_1064_ are the new RGB images at the wavelength of 780 nm and 1064 nm, respectively, and M is a mask matrix that represents the threshold when overlaying the 780 nm image onto the 1064 nm image (A_780_ > 0, M = 1, else M = 0). **·** × represents the pixel-wise multiple. 

## 3. Results

### 3.1. Characterization of PA Performances of ICG-Lips

The PA performances of the ICG-Lips were evaluated by our PACT system. ICG and ICG-Lips were diluted to the same concentration (50 µg/mL) in phosphate-buffered saline (PBS). Then, the PBS, ICG, and ICG-Lips were separately injected into three transparent plastic tubes and captured by PACT with the laser wavelength 780 nm. As shown in Figure 7, the PA performance between ICG and ICG-Lips was not significantly different. The prepared ICG-Lips retains the PA properties of ICG.

### 3.2. Imaging of the ICG-Lips in Tumors

The accumulation of ICG-Lips in tumor was monitored by our PACT system. Firstly, the mouse was anesthetized and placed vertically in the center of the transducer for cross-sectional imaging before injecting the ICG-Lips (Figure 1b and Figure 8a). Then, the ICG-Lips was injected into mice via tail vein at a volume of 100 µL [23]. The PA signal was captured for ~24 h. As shown in Figure 8, where the surrounding tissues are shown in gray captured with 1064 nm laser wavelength, and the PA effects of 780 nm in tumor are highlighted in color. After 5 min of injection, there was a significant enhancement of the PA signal in the ROI. The accumulation ICG-Lips via enhanced permeability and retention (EPR) effects lasted about 90 min. 

For long-term monitoring, respiration in the mouse caused a slight drifting during the imaging process. To overcome this problem, we acquired 30 frames of data at each time point and drawn a line graph for the trend of the 780 nm PA signal of the ROI at different time points. As Figure 9 shows, the PA signal had a significant change from pre-injection to 24 h. After 5 min of ICG-Lips injection, the PA amplitude at the tumor site began to show an increasing trend. After 90 min of ICG-Lips injection, the PA amplitude of the ROI had dropped. From the results and the images, ICG-Lips in the tumor is abundantly aggregated within 30 min to 90 min after injection. After that, the PA amplitude decreased, indicating that ICG was gradually cleared by the liver. This result guided the best time for phototherapy to be 30 min to 90 min after the injection. 

The results indicated that our PACT system could monitor the dynamic biodistribution of ICG-Lips in tumors in real time and provide treatment guidance for phototherapeutic applications.

## 4. Discussion

With the high spatial resolution and rich optical contrast, our PACT system has the ability to estimate in vivo PA performance of different photoacoustic contract agents. Here, ICG-encapsulated liposomes are prepared as PA imaging agents. In vitro PA studies indicated that ICG-Lips and free ICG had compatible PA effects. (Figure 7e). Moreover, due to the high sensitivity of our PACT system, the in vivo PA effects of ICG-Lips can be captured by our platform in real-time. Our dual-wavelength PACT system, integrating the real-time imaging of drug distribution and surrounding tissues, provides vital guidance for phototherapeutic applications. 

Although the dual-wavelength PACT system has shown its great potential in disease treatment, its clinical applications for humans still need further development. For example, tissue penetration issues still hinter the imaging quality of PACT. We will also continue to explore new ways to improve system performance. 

In summary, we report a dual-wavelength PACT system to monitor the dynamic bio-distribution of ICG-Lips in tumors. Owing to the non-invasiveness and high sensitivity, our PACT system offers a promising imaging platform for phototherapeutic guidance. 

## Figures and Tables

**Figure 1 micromachines-13-00946-f001:**
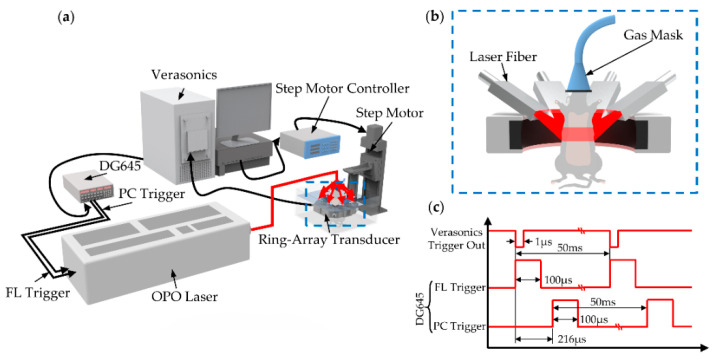
System structure schematic and trigger timing diagram. (**a**) Schematic of the dual-wavelength PACT system for tumor imaging; (**b**) Close up of the blue dashed box region in (**a**), which shows the distribution of the laser fibers and the placement of mouse; (**c**) System trigger timing diagram.

**Figure 2 micromachines-13-00946-f002:**
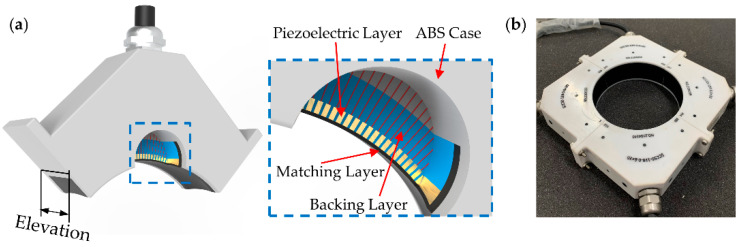
(**a**) The configuration of the transducer. The blue dotted box is a close-up of the transducer; (**b**) The photograph of the ring-array transducer.

**Figure 3 micromachines-13-00946-f003:**
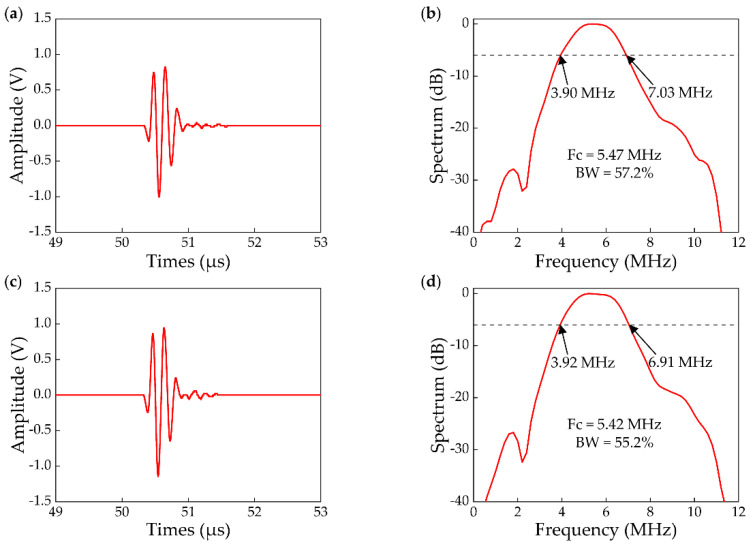
Transmit characteristic of the ring-array transducer. (**a**,**b**) Measured waveform and spectrum diagram of the 25-th element of the ring-array transducer; (**c**,**d**) Measured waveform and spectrum diagram of the 96-th element of the ring-array transducer.

**Figure 4 micromachines-13-00946-f004:**
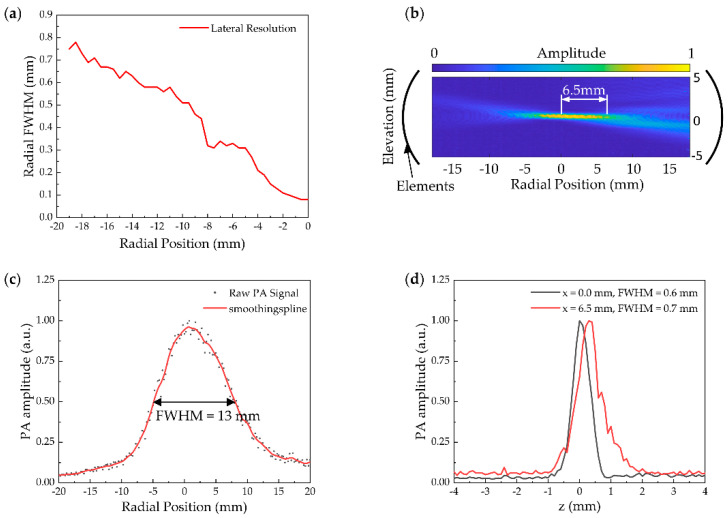
(**a**) Experimental cross-sectional view of the field of view (FOV) with location for a radical cut dividing the transducer into two halves; (**b**) Measured lateral resolution versus distance from the center of the transducer; (**c**) The full width at half maximum (FWHM) of (**a**) the radial FOV focus area; (**d**) The FWHM of (**a**) at the center of the ring-array (x = 0 mm) and at 6.5 mm in the elevation direction.

**Figure 5 micromachines-13-00946-f005:**
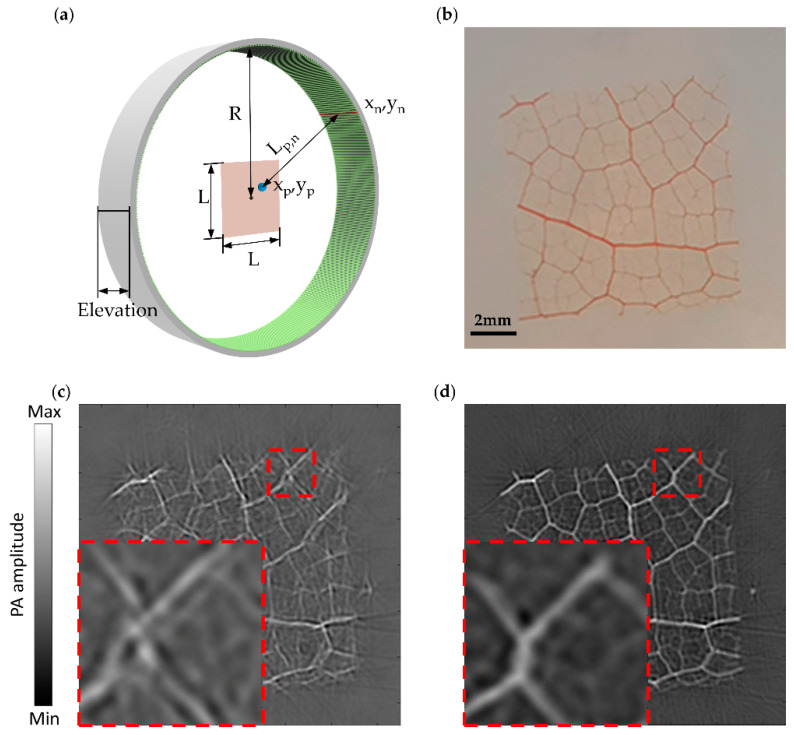
Theoretical analysis model and the comparison of results before and after transducer geometry correction. (**a**) The definition of both the array element coordinates and the FOV of the image reconstruction area. (x_n_, y_n_) represents the position of the n-th array element and (x_p_, y_p_) represents the pixel location of the point source. L is the size of the image reconstruction area. L_p,n_ is the distance between the array element (x_n_, y_n_) and the pixel point (x_p_, y_p_); (**b**) The sample of leaf vein for imaging; (**c**,**d**) The PA image of the leaf vein before and after transducer geometry correction. The red dotted box is a close-up of the same area of the leaf veins.

**Figure 6 micromachines-13-00946-f006:**
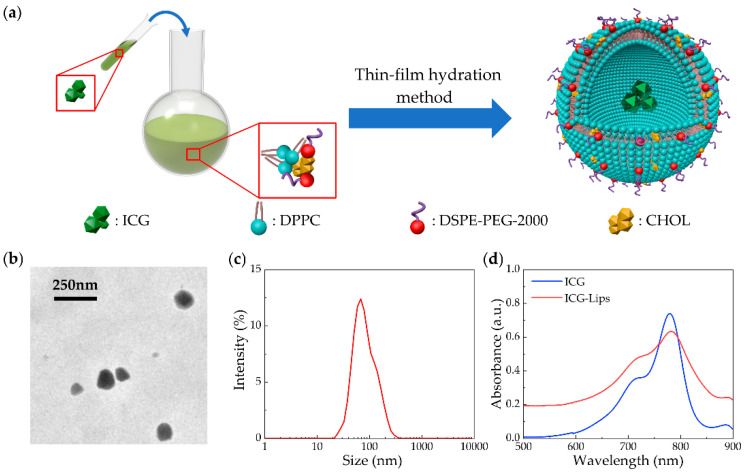
Schematic of the preparation process and the characterization of ICG-Lips. (**a**) ICG-Lips are generated by the thin-film hydration method. ICG, indocyanine green. DPPC, dipalmitoyl phosphatidylcholine. DSPE-PEG, 1,2-distearoyl-sn-glycero-3-phosphoethanolamine-N-[amino (polyethylene glycol) 2000]. CHOL, cholesterol; (**b**) Transmission electron microscope (TEM) image of ICG-Lips; (**c**) Size distribution of ICG-Lips (z-average 129 ± 49 nm); (**d**) UV-Vis spectra of ICG and ICG-Lips in phosphate-buffered saline.

**Figure 7 micromachines-13-00946-f007:**
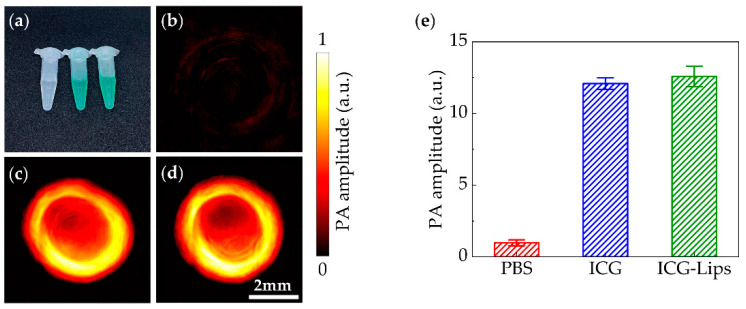
The comparison of PA performances. (**a**) The photograph of PBS, free ICG, and ICG-Lips (containing 50 µg/mL ICG) in centrifuge tubes; (**b**–**d**) PA images of PBS, free ICG, and ICG-Lips; (**e**) The PA amplitude of PBS, free ICG, and ICG-Lips in transparent plastic tubes with laser wave-length at 780 nm. PBS, phosphate-buffered saline. Error bars represent the standard deviations from 30 independent measurements.

**Figure 8 micromachines-13-00946-f008:**
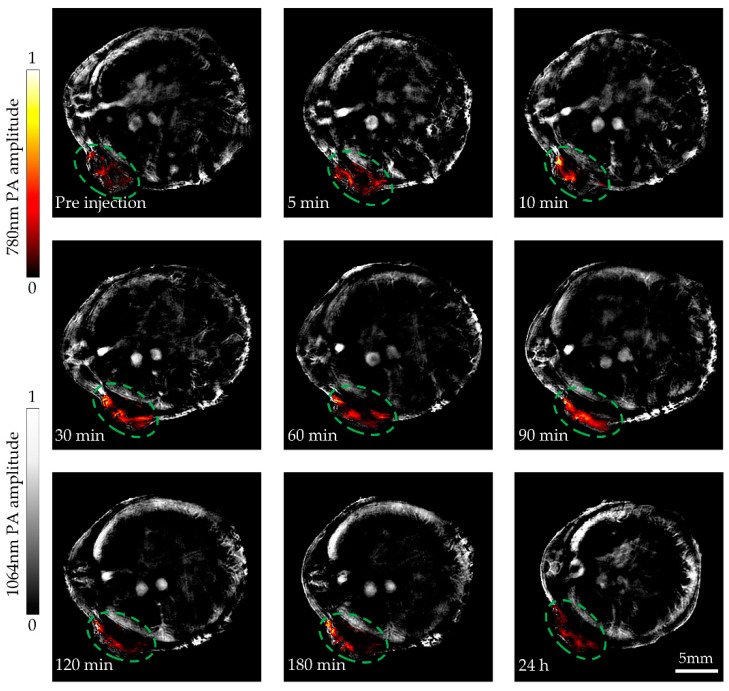
ICG-Lips accumulation in tumors for 24 h was monitored by our PACT system. The PA images of mouse tissues imaged by a 1064 nm wavelength laser are shown in gray. The PA images of ICG-Lips in tumors imaged by a 780 nm wavelength laser are shown in color.

**Figure 9 micromachines-13-00946-f009:**
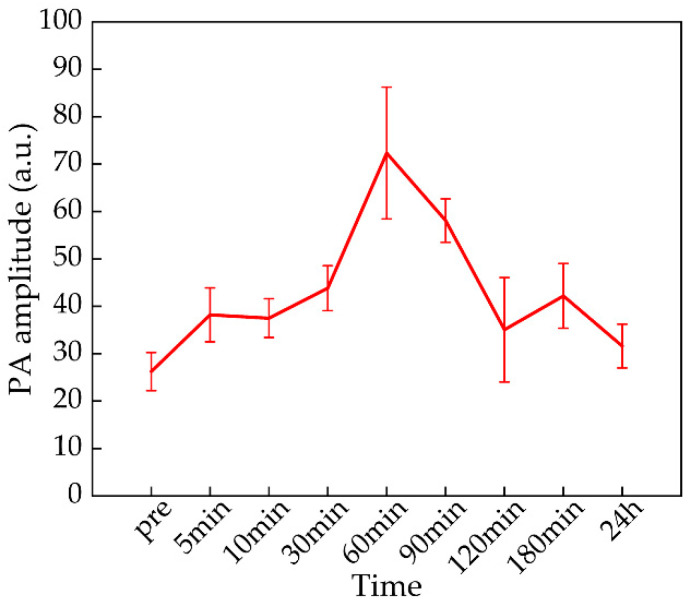
780 nm PA amplitude of the ROI at different time points. Error bars represent the standard deviations from 30 independent measurements.

**Table 1 micromachines-13-00946-t001:** The design parameters of the ring-array transducer.

Layer	Sound Velocity(m/s)	Density (kg/m^3^)	Acoustic Impedance (×106 kg/(m^2^·s))	Thickness (mm)
Matching layer	2635	1180	3.11	0.13
Backing layer	1730	2050	3.55	7
Piezoelectric layer	3737	3480	13.00	0.3

**Table 2 micromachines-13-00946-t002:** The properties of the piezoelectric layer.

Material	ε^τ^_33_ ^1^ (1 kHz)	d_33_ ^2^ (Pc/n)	k_t_ ^3^	Q_m_ ^4^	Dielectric Loss (1 kHz (tan δ))	ƒ_t_ ^5^ (MHz)	ƒ_α_ ^6^ (MHz)
PZT/epoxy 1–3 composite	1361.8	493	0.612	18	0.028	5.006	6.107

^1^ Dielectric constant; ^2^ Piezoelectric strain constant; ^3^ Electromechanical coupling coefficient; ^4^ Mechanic quality factor; ^5^ Resonant frequency; ^6^ Anti-resonant frequency.

## Data Availability

The data presented in this study are available on request from the corresponding author. The data are not publicly available due to the confidentiality of the relevant codes.

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
