# Peer review of "Dual-Wavelength Photoacoustic Computed Tomography with Piezoelectric Ring-Array Transducer for Imaging of Indocyanine Green Liposomes Aggregation in Tumors"

_micromachines, 2022, doi:10.3390/mi13060946_

Round 1
Reviewer 1 Report
In this study, the authors proposed a dual-wavelength photoacoustic computed tomography (PACT) system, which has a piezoelectric ring-array transducer to image the aggregation of ICG-Lips in tumors so as to guide phototherapy. A visible 780 nm light excites the photoacoustic (PA) effects of the ICG-Lips and a near-infrared 1064 nm light provides the imaging of the surrounding tissues. The aggregation of ICG-Lips within the tumor and the surrounding tissues can be visualized by the PACT continuously. The results indicated that the PACT system can realize a real-time monitoring for ICG-Lips accumulation in tumors.
To my understanding, Fig. 8 is an improtant proof for the effectiveness of the PACT system. Can the authors give a detailed explanation about it? With the time passing, does the ICG-Lips accumulation in tumors have any difference?
Author Response
Dear Editor and reviewers,
We have complied with the reviewers’ comments.
Please see the attachment

Reviewer 2 Report
The manuscript describes a promising PACT system with a piezoelectric ring-array transducer to monitor the in vivo ICG-liposome distribution in tumors. The manuscript is well structured and clearly written. However, this manuscript still needs some improvements before being published. The comments are listed as follows:
1. In section 2.2, DG645 generated two triggers (FL and PC trigger) with a difference of 216 µs. Please explain why the delay is 216 µs?
2. Figure 2a should be improved. More details of the transducer should be added.
3. Figure 4c is not clear to show that the FWHM of the FOV section is 13 mm, and the title of this is misspelled. What is "posation" here?
4. In Figure 5, the pre-injection image also shows some 780nm PA signal, which may be due to the tumor tissue absorption at 780nm. For the following ICG injection images, is it possible to subtract this 780nm background signal, to only show the ICG contribution?
5. In Table 1, the numbers in the units of density and speed of sound need to be superscripted.
6. For long-term monitoring of ICG dynamics in tumors, it is essential to make sure the PACT images are taken from the exact same position of the mouse at different time points. The anatomy structure in the 180min image is slightly different from that in the 5min image. It seems the imaging plane has a slight drifting during the imaging process. How does it affect the quantitative analysis of the ICG dynamics?
7. In Figure 8, the PA signals of ICG-Lips at 120 min time point showed a declining trend. However, the PA signals at 180 min point were higher. The authors are advised to check the results and demonstrate the reasons. Furthermore, the figure includes two PA amplitude colormaps. For clarity, it is recommended to name them as '780nm PA amplitude' and '1064 PA amplitude' respectively.
8. English still needs to be polished.
Author Response

(The authors gave the same response as above.)

Reviewer 3 Report
In this manuscript, a dual-wavelength PAT was developed to image and monitor ICP-Liposomes aggregation in tumors. The PAT system was made of four quarter-ring array transducer and calibrated before being used in an animal study. The phantom study validated the imaging system and the imaging approach. The further animal study confirmed that the aggregation of ICP-Liposomes in the tumor can be imaged at different time points after injection. This is a very interesting study, and this manuscript is publishable after addressing the following questions.
1) What reconstruction algorithm was used in this study? Provide more technical details of the reconstruction algorithm used.
2) What is the temporal resolution of the developed PAT approach (data acquis ion time, reconstruction time, etc.)?
Author Response

(The authors gave the same response as above.)
